Exploring freshwater soundscapes of tropical marshland habitats in Southeast Asia: insights into auditory sensory adaptation of wild Siamese fighting fish Betta splendens

Ramos Andreia 1
Gonçalves David 1
Vasconcelos Raquel O. raquel.vasconcelos@usj.edu.mo 1 2 3
1 Institute of Science and Environment, University of Saint Joseph , Macao , Macao S.A.R. , China
2 MARE–Marine and Environmental Sciences Centre / ARNET–Aquatic Research Network, Faculdade de Ciências, Universidade de Lisboa , Lisbon , Portugal
3 EPCV–Department of Life Sciences, Lusófona University , Lisbon , Portugal
Esteban María Ángeles
Electronic publication date: 2025 Jan 13
Publication date: 2025
Volume: 13
Electronic Location ID: e18491
Received 2024 Aug 8; Accepted 2024 Oct 17
Copyright: ©2025 Ramos et al.
Copyright year: 2025
Copyright holder: Ramos et al.
License: This is an open access article distributed under the terms of the Creative Commons Attribution License, which permits unrestricted use, distribution, reproduction and adaptation in any medium and for any purpose provided that it is properly attributed. For attribution, the original author(s), title, publication source (PeerJ) and either DOI or URL of the article must be cited.
License URL: https://creativecommons.org/licenses/by/4.0/

Keywords: Ambient noise, Freshwater acoustics, Hearing thresholds, Auditory evoked potentials, Siamese fighting fish

Funding: Science and Technology Development Fund (FDCT), Macau 0068/2020/A2 Postdoctoral Research Funding 0005/2021/APD This study was supported by the Science and Technology Development Fund (FDCT), Macau, through the project ref. 0068/2020/A2 and Postdoctoral Research Funding for AR (ref. 0005/2021/APD). The funders had no role in study design, data collection and analysis, decision to publish, or preparation of the manuscript.

==============================
While soundscapes shape the structure and function of auditory systems over evolutionary timescales, there is limited information regarding the adaptation of wild fish populations to their natural acoustic environments. This is particularly relevant for freshwater ecosystems, which are extremely diverse and face escalating pressures from human activities and associated noise pollution. The Siamese fighting fish Betta splendens is one of the most important cultured species in the global ornamental fish market and is increasingly recognized as a model organism for genetics and behavioural studies. This air-breathing species (Anabantoidei), characterized by the presence of a suprabranchial labyrinth organ that enhances auditory sensitivity, is native to Southeast Asia and inhabits low flow freshwater ecosystems that are increasingly threatened due to habitat destruction and pollution. We characterized the underwater soundscape, along with various ecological parameters, across five marshland habitats of B. splendens, from lentic waterbodies to small canals near a lake in Chiang Rai province (Thailand). All habitats exhibited common traits of low dissolved oxygen and dense herbaceous vegetation. Soundscapes were relatively quiet with Sound Pressure Level (SPL) around 102–105 dB re 1 µPa and most spectral energy below 1,000 Hz. Sound recordings captured diverse biological sounds, including potential fish vocalizations, but primarily insect sounds. Hearing thresholds were determined using auditory evoked potential (AEP) recordings, revealing best hearing range within 100–400 Hz. Males exhibited lower hearing thresholds than females at 400 and 600 Hz. This low-frequency tuning highlights the potential susceptibility of B. splendens to anthropogenic noise activities. This study provides first characterization of the auditory sensitivity and natural soundscape of B. splendens, establishing an important ground for future hearing research in this species. The information provided on the auditory sensory adaptation of B. splendens emphasizes the importance of preserving quiet soundscapes from lentic freshwater ecosystems.

Introduction

Although freshwater ecosystems occupy a small portion of the Earth’s surface (0.8%) and just 0.01% of the total water on Earth, these habitats are home to nearly 43% of all modern fish (Nakatani et al., 2011; Nelson, Grande & Wilson, 2016). Freshwater fishes are also known to play a key role in controlling the trophic structure of their ecosystems (Villéger et al., 2017; Su et al., 2021). While this remarkable biodiversity has been continuously documented, the diversification of freshwater fishes on a global scale and the environmental pressures driving their speciation remain largely unexplored (Manel et al., 2020; Cerezer et al., 2023).

Freshwater habitats have traditionally been characterized in terms of physical, chemical, and hydrological parameters (Thomson et al., 2001; Belmar et al., 2019; Picado et al., 2020) but very limited information exists on their acoustic features (Tonolla et al., 2010; Desjonquères et al., 2015; Lara & Vasconcelos, 2019; Rountree, Juanes & Bolgan, 2020; Decker et al., 2020). While several studies focused on the biological sounds produced by soniferous fish species (Anderson, Rountree & Juanes, 2008; Montie, Vega & Powell, 2015), some researchers described other biophonies such as insect sounds (Desjonquères et al., 2020) and acoustic patterns associated to geophysical processes such as hydraulic turbulence or sediment transport (Tonolla et al., 2011).

Underwater soundscapes can exhibit remarkable complexity, often surpassing terrestrial acoustic environments in richness of information (Fay, 2009). This understanding has become increasingly evident as fish species demonstrate the capability to detect and process both sound pressure and particle motion, as well as to perform auditory scene analysis and sound source localization (Hawkins & Popper, 2018; Popper & Hawkins, 2019; Popper & Hawkins, 2021; Veith et al., 2024). The limitations of other sensory channels such as vision, olfaction, and touch in the aquatic environment, particularly in providing rapid, long-distance, and tridimensional information, make sound an exceptionally efficient information carrier for most aquatic organisms (Popper & Hawkins, 2019; Popper & Hawkins, 2021).

By listening to the aquatic environment, fish can extract key biotic cues concerning the presence of conspecifics and heterospecifics, including potential mates, prey, and predators (Putland, Montgomery & Radford, 2019). Additionally, they may detect abiotic information for orientation, such as sounds from wind, water currents, turbulence, and moving substrate (e.g., Lagardere et al., 1994).

Many fish species that live in freshwater environments exhibit higher auditory sensitivity and/or an extended frequency bandwidth, due to accessory structures that enhance hearing by acoustically coupling air-filled cavities to the inner ear (Ladich, 2000). Examples of such structures are the Weberian apparatus found in Ostariophysi and the labyrinth organ in Anabantoidei, which allows them to survive in low-oxygen stagnant waters (Yan, 1998; Yan et al., 2000; Ladich & Yan, 1998; Ladich & Popper, 2004). The labyrinth organ is directly connected to the inner ear, enabling sound pressure detection, and enhancing their hearing sensitivity and detectable frequency range (Schneider, 1942; Saidel & Popper, 1987; Yan, 1998; Ladich & Yan, 1998). The sensory adaptation of these fishes to their highly diverse acoustic habitats, particularly their ability to extract relevant biological information, remains poorly investigated. Few studies suggested that acoustic communication was not the primary selective pressure involved in the evolution of their auditory abilities (Ladich, 2000), emphasizing that the detection of acoustic cues from sources other than conspecifics might have been crucial. This lack of information calls for further research on hearing specialists, such as otophysan and anabantoid species, which predominately inhabit freshwater ecosystems that are increasingly threatened by habitat destruction and pollution (Gozlan et al., 2019; Rountree, Juanes & Bolgan, 2020).

The Siamese fighting fish Betta splendens (Anabantoidei: Osphronemidae) is one of the most important cultured species in the global ornamental fish market, having undergone extensive selective breeding for distinctive physical traits and behaviors (Simpson, 1968; Kwon et al., 2022). Wild-type populations of this species persist in regions of Southeast Asia, including Thailand, the northern Malay Peninsula, Cambodia, Indonesia and southern Vietnam, where they inhabit poorly oxygenated freshwater environments such as small ponds and rice paddies (Hui & Ng, 2005; Kowasupat, Panijpan & Ruenwongsa, 2012; Panijpan et al., 2017). As an anabantoid fish, B. splendens has developed a highly folded and vascularized accessory breathing organ from the epibranchial bone, termed the labyrinth organ (Adamek-Urbańska et al., 2021). This apparatus allows extracting oxygen from air, equipping anabantoids to persist in hypoxic and polluted water (Tate et al., 2017).

Several studies on Siamese fighting fish, using both wild and domesticated strains, have shown the potential of using this species for ecotoxicology (Tudor et al., 2019) and for decoding the genomic and physiological mechanisms of behavior, particularly aggression (Fan et al., 2018; Ramos & Gonçalves, 2019; Kwon et al., 2022; Ramos & Gonçalves, 2022; Palmiotti et al., 2023). However, there are very few studies focusing on the species’ natural habitat and no information exists on its natural soundscape and auditory sensitivity.

In this study, we characterized the underwater soundscape and various ecological parameters across different marshland habitats of B. splendens, ranging from lentic waterbodies to small canals, in Chiang Rai province, northern Thailand. Additionally, we assessed the auditory sensitivity of a wild strain originated from the same region, including males and females, to evaluate the species’ sensory adaptation to its natural acoustic environment and compare it with a sympatric vocal anabantoid species.

Methods

Portions of this text were previously published as part of a preprint (Ramos, Gonçalves & Vasconcelos, 2024).

Test animals

The Siamese fighting fish (B. splendens) used for audiometry purposes were 8 months old, consisting of five males (31.98 ± 5.9 mm standard length, mean ± standard deviation, 27.40–37.40 range) and 6 females (29.65 ± 3.97 mm, 22.70–33.70 mm). These fish belonged to the 10th generation, originating from adult fish initially provided by suppliers in Chiang Rai, Thailand. They were bred and reared at the fish facility of the University of Saint Joseph (Macao), following previously described procedures (Ramos & Gonçalves, 2019; Ramos & Gonçalves, 2022; Lichak et al., 2022). The fish tanks were maintained at 28 ± 1 °C and under 12:12 light-dark photoperiod. All animals were fed twice daily with a mixture of dry food (pellets from various brands) and live artemia. Fish health status was monitored daily based on their feeding activity, swimming behaviour, and the absence of apparent diseases.

Test subjects were initially maintained in mixed-sex stock tanks (50 cm width × 30 cm length × 25 cm height) equipped with external filtering system and enriched with aquatic vegetation and small ceramic shelters. Each tank housed a maximum of 50 fish, and presented a background sound pressure level (SPL) of 130–136 dB re 1 µPa (based on LZS, i.e., RMS sound level obtained with slow time and linear frequency weightings; flat weighting: 6.3 Hz–20 kHz; see measuring equipment described below). Prior to audiometry, fish were isolated in small individual tanks (28 cm width × 14 cm length × 20 cm height; 4.5 L) for one week under lab silent conditions with SPL varying between 103 and 108 dB re 1 µPa. Both male and female specimens were tested in both the morning and afternoon periods, on alternating days. After the audiometry tests, the animals recovered in isolated tanks with aerated system water and were later returned to the fish facility. These fish remained in isolated tanks and were only used in further experiments at least 1 month later.

All experimental procedures complied with the ethical guidelines enforced at the University of Saint Joseph and approved by the Division of Animal Control and Inspection of the Civic and Municipal Affairs Bureau of Macao, license AL017/DICV/SIS/2016.

Field sound recordings

Field sound recordings were performed away from urban areas in various locations within marshland habitats in Chiang Rai (Northern Thailand), where the study species had been observed in previous studies (Ramos & Gonçalves, 2019; Ramos & Gonçalves, 2022). These marshlands were modified for agricultural purposes, featuring stagnate pools, lentic canals, and drainage dikes to manage the water supply. Ambient sound recordings and SPL measurements were carried in five distinct locations near the Chiang Saen Lake, Chiang Saen District (CS) and the Rai Son Thon Reservoir (RST), Wiang Chai District (Fig. 1, Table 1): CS1–drainage canal bordered by tall dense vegetation; CS2–water body within an extensive mudflat with short herbaceous vegetation; RST1–canal surrounded by dense vegetation with presence of water buffalos; RST2 and RST3–water bodies in marshland with diverse herbaceous plants, including grasses, sedges, rushes, and other aquatic plants. Overall, these freshwater habitats exhibited variable turbidity, brownish-coloured water and muddy clay substrate, similarly to previously described habitats for the study species (Nur et al., 2022).

Figure 1 Geographical locations of the different study sites in Northern Thailand, Chiang Rai Province, where wild Siamese fighting fish (Betta splendens) were identified and sound recordings conducted.

(Top row) Grey map highlights the two major locations: Chiang Saen District (CS) and the Rai Son Thon Reservoir (RST), Wiang Chai District. Satellite images (Google Earth) of the respective locations are shown to provide topographical details. (Bottom row) Images from each sampling site during recordings: CS1–drainage canal bordered by tall, dense vegetation; CS2–water body within an extensive mudflat with short herbaceous vegetation; RST1–canal surrounded by dense vegetation with the presence of water buffalos; RST2 and RST3–water bodies in marshland with diverse herbaceous plants. Each image shows the hydrophone attached to a rod fixed to the bottom (white arrows). An image captured by an underwater camera in one of the study sites shows water visibility (top right) and a bubble nest (bottom right) are also presented. All pictures were taken by A. Ramos and D. Gonçalves, coauthors of this work, and are therefore free of copyrighted material.

An initial assessment of the water physico-chemical properties was carried out using a handheld multi parameter sensor (Pro Plus; YSI, Yellow Springs, OH, USA) for each of the study locations. The sound recording protocol followed previous procedures by Lara & Vasconcelos (2019). The ambient noise and SPL were recorded using a hydrophone (Brüel & Kjær 8104, Naerum, Denmark; frequency range: 0.1 Hz–120 kHz, sensitivity of −205 dB re 1 V/µPa) connected to a hand-held sound level meter with sound recording function (Brüel & Kjær 2250). The hydrophone was attached to a pole and positioned in the middle of the water column avoiding direct contact with substrate and vegetation.

Sound recordings, with a sampling frequency of 48 kHz, consisted of two consecutive 15 min sessions per site. The equivalent continuous SPL (LZeq, linear broadband spectrum or flat weighting (z): 6.3 Hz–20 kHz) averaged over 60 s was obtained six times per site - three times immediately before and after the overall 30 min recording. LZeq (also known as LLeq) is a measure of averaged energy in a varying sound field and is commonly used in environmental noise studies (ISO 1996 2003). The field work was conducted in March 13–14th 2023, between 10:00–17:00 h, under tropical dry season conditions and without rainfall.

Table 1 Characterization of the study sites within the marshland habitats of B. splendens at Chiang Saen Lake, Chiang Saen District (CS1 and CS2) and the Rai Son Thon Reservoir, Wiang Chai District (RST1, RST2 and RST3). DO, Dissolved Oxygen.

Location	GPS coordinates	Habitat	Elevation (m)	Water temp. (° C)	Depth (m)	Salinity (ppt)	Conductivity (µs/cm)	pH	DO (mg/L)	
CS1	20°15′51.14″N100°2′46.67″E	Drainage canal bordered by tall vegetation	371	22.5	0.30	0.04	81.3–81.5	5.6–5.7	0.19–0.37	
CS2	20°15′58.46″N100°2′33.03″E	Water body within extensive mudflat with short herbaceous plants	369	24.4–25.1	0.16	0.01	33.2–33.5	5.4–5.5	2.87–3.02	
RST1	19°49′27.56″N99°59′10.39″E	Canal with dense vegetation and presence of water buffalos	391	24.4	0.20	0.04	92.8	5.8	0.40	
RST2	19°50′5.19″N100°0′12.77″E	Water body within mudflat surrounded by herbaceous plants	393	22.1	1.20	0.03	53.5–54.2	4.8–4.9	0.38–0.64	
RST3	19°50′6.55″N100°0′11.22″E	Water body within mudflat surrounded by herbaceous plants	393	22.6	0.76	0.02	47.1–59.8	5.6–5.6	0.61–0.64	

Sound analysis

Sound recordings were initially analyzed using Raven 1.5 (Bioacoustic Research Program, Cornell Laboratory of Ornithology, NY, USA). Both aural and visual inspection of all sound recordings were applied to identify potential sound sources.

Spectral analysis was performed using Adobe Audition 3.0 (Adobe Systems Inc., San José, CA, USA) based on the overall 30 min per site. The Power Spectral Density (PSD) level, expressed in dB re 1 µPa2/Hz, as well as the absolute sound spectra level in dB re 1 µPa, were determined by using the averaged LZeq values calculated for each site, in accordance with the methods outlined in prior research (Amoser & Ladich, 2010; Lara & Vasconcelos, 2019). The PSD level was calculated using the following process: first, the linear spectral amplitude Ai was derived from the logarithmic spectral amplitude ai using the equation Ai = 10 (ai/10). Subsequently, these linear amplitude values were converted to PSD levels through the equation: PSD level (dB) = 10 × log10AiBW2, where BW is the bandwidth (spectral resolution). Spectrograms were generated using a 16386-point Fast Fourier Transform (FFT) with a Hanning window.

To compare the spectral profiles of natural soundscapes with the audiogram and sound spectra of a soniferous fish species observed in the study locations, we analysed a sound recording of the croaking gourami Trichopsis vittata (obtained at 25 °C), provided by F. Ladich, along with previously published auditory thresholds for this species (Wysocki & Ladich, 2001). This allowed us to compare auditory sensitivities between two sympatric species from the same family (Osphronemidae), which display contrasting behavioural strategies (vocal vs. non-vocal) and potentially different sensory adaptations.

Auditory evoked potential measurements

The Auditory Evoked Potential (AEP) recording protocol was according to prior studies (Breitzler et al., 2020; Wong et al., 2022). Audiometry tests were conducted in a rectangular plastic tank measuring 50 cm length × 35 cm width, and 23 cm height, and filled with system water at 28 ± 1 °C. An underwater speaker (UW30, Electro-Voice, MN, USA) was placed at the bottom of the tank and surrounded by a layer of fine sand. A custom-designed acoustic stimulation system was used for low frequencies (100 Hz), featuring a vibrating plexiglass disc operated by a mini shaker (Brüel & Kjær 4810). This setup was mounted at the center of the front-facing tank wall. The entire experimental arrangement was placed on an anti-vibration air table (Vibraplane, KS Kinetic Systems, MA, USA) within a walk-in soundproof chamber (120a-3, IAC Acoustics, North Aurora, IL, USA).

Adult B. splendens were initially subjected to a light anaesthesia bath using 400 mg/L tricaine methanesulfonate (MS-222, Arcos Organics, NJ, USA), buffered with double concentration of sodium bicarbonate, until opercular movement ceased, which took approximately 3–5 min. The fish were placed in a specially designed sponge holder that immobilized them, with a net partially covering the upper body, ensuring the fish’s head remained just below the water surface to allow normal breathing after recovering from anaesthesia. Two stainless steel electrodes (0.40 mm diameter, 13 mm length, Rochester Electro-Medical, Inc., FL, USA) were utilized, with the recording electrode securely positioned against the skin over the brainstem area and the reference electrode placed on the side of the body (Fig. 2C).

Figure 2 Examples of auditory evoked potential (AEP) recordings of B. splendens.

Examples of auditory evoked potential (AEP) response curves of two B. splendens males in response to (A) 400 Hz and (B) 800 Hz tone stimuli presented at decreasing amplitudes (dB re 1 µPa of each tone stimuli are indicated in front of corresponding AEP curve). Auditory tone stimuli are shown on top (red) for the two frequencies. Auditory thresholds (indicated by asterisks) were identified as the lowest amplitude at which a consistent response curve pattern could be observed. Voltage scales were adjusted to enhance visibility and correspond to the AEP curves presented below. (C) Test specimen positioned on a sponge holder in the AEP recordings setup.

Sound stimuli and AEP recordings were controlled via a TDT audiometry workstation (Tucker-Davis Technologies, FL, USA). The AEP signal was transmitted into a low-impedance head stage (RA4LI, TDT) that was connected to a pre-amplifier (RA4PA, TDT), and then band-pass filtered (0.1–1 kHz) and digitized (16 bit, ± 4 mV). The output was finally sent to a Multi-I/O processor (RZ6, TDT). While SigGen software allowed generation of sound stimuli, BioSig TDT software controlled both signal presentation and acquisition. Stimuli consisted of 20 ms tone bursts of 100, 200, 400, 600, 800, 1,000, 2,000, 4,000 and 6,000 Hz, including 2 ms rise/fall ramps. Tone stimuli were presented randomly at least 1,000 times, half at opposite polarities (180° phase shifted). The system was calibrated before each audiometry session by placing a hydrophone (Brüel & Kjær 8104) connected to a sound level meter (Brüel & Kjær 2270) to verify the SPL at the position occupied by the test subjects.

For each frequency, tones were initially presented at 140 dB re 1 µPa and then following 2.5 dB consecutive decrements. The auditory threshold referred to the minimum level required to elicit an identifiable and reproducible AEP response in at least two averaged waveforms. To confirm an auditory response, at least two of the following criteria were reached: (a) matching waveform shape compared to previous sound level; (b) increased latency in a consistent AEP peak with decreasing sound level; and/or (c) spectral peak in the FFT analysis of the AEP response that is twice the stimulation frequency (Figs. 2A, 2B).

Statistical analysis

The overall noise levels (LZeq) from the recording sites were compared using one-way ANOVA, followed by post hoc Tukey tests to verify habitat specific differences.

Auditory thresholds at different frequencies from males and females were compared with two-way repeated measures GLM with contrast analysis, with sex as a between-subject factor and the different sound frequencies as repeated measures (within-subject factor). Parametric assumptions were complied, namely data were normally distributed and variances were homogeneous. The statistical analysis was performed with IBM SPSS v26 (IBM Corp. Armonk, NY, USA).

Results

Characterization of natural habitats

The Siamese fighting fish B. splendens were identified in different marshland habitats with dense herbaceous vegetation, from lentic waterbodies in the mudflat to small canals near a lake. All freshwater habitats exhibited common traits of low dissolved oxygen, from 0.19–0.37 mg/L (CS1) in a small lentic canal to 2.87–3.02 mg/L in a small pool (CS2), with variable turbidity–see Table 1, Fig. 1. Most of the habitats were very shallow, below 1 m (0.2–0.76 m), and only one location (RST2) was slightly deeper (1.2 m). The overall water temperature varied between 22.1 and 25.1 °C and presented low conductivity levels (<92.8 µS/cm).

All five study sites consisted of quiet locations with similar noise levels (SPL or L Zeq) and low variability within each site (CV < 3%). Overall, SPL varied between 101.6 ± 0.1 and 105.4 ± 2.9 dB re 1 µPa (mean ± standard deviation) (Table 2, Figs. 3 and 4). Nevertheless, significant differences in SPL were found between recording sites (F (4, 29) = 8.225; p < 0.001), with RST2 showing higher levels compared to the other sites (post hoc tests, p ≤ 0.003) probably due to the constant presence of insect sounds (Figs. 3 and 4).

Table 2 Mean sound pressure levels (dB re 1 µPa) ± standard deviation (SD) measured at five study sites at Chiang Saen Lake, Chiang Saen District (CS1 and CS2) and the Rai Son Thon Reservoir, Wiang Chai District (RST1, RST2 and RST3).

Mean values were determined from six readings based on 60 s; CV, coefficient of variation.

Location	Mean ± SD	Min	Max	CV (%)	
CS1	102.2 ± 0.5	102.2	105.5	0.53	
CS2	102.2 ± 0.3	102.1	104.0	0.32	
RST1	101.6 ± 0.1	101.8	104.7	0.12	
RST2	105.4 ± 2.9	104.5	108.6	2.78	
RST3	101.8 ± 0.2	101.9	104.2	0.16	

Figure 3 Characterization of ambient noise from diverse marshland habitats recorded in Chiang Saen District (CS) and Rai Son Thon Reservoir (RST).

(A) Power spectral density (PSD) of ambient noise from diverse marshland habitats recorded in Chiang Saen District (CS) and Rai Son Thon Reservoir (RST) in March 13–14th, 2023. Sampling frequency: 48 kHz, FFT size 16384. (B) Spectrogram from RST2 presenting high-pitched insect calls (between 10–15 kHz) and potential fish sounds—marked (white square) and detailed in (C); (D) and (E) oscillogram and spectrogram of cavitation sounds (from RST2) and possible fish sound (from CS1), respectively. FFT size 2048.

Figure 4 Comparison of mean Sound Pressure Level (±standard deviation) between different recording sites.

Measurements were carried in Chiang Saen District (CS) and Rai Son Thon Reservoir (RST) (F (4, 29) = 8.225; p < 0.001). Each averaged value was based on six measurements (linear equivalent, LZeq) per site over 60 s period. Distinct letters indicate statistically significant differences as determined by pairwise post hoc comparisons.

Soundscapes from RST locations revealed distinct biological sounds, including potential fish vocalizations consisting of series of pulsed sounds <2,000 Hz (RST2), birds singing at distance with main energy between 1,000–7,000 Hz (RST2), but primarily insect sounds >2,000 Hz, especially in RST2 (Fig. 3). The soundscapes of both CS1 and CS2 did not reveal much underwater biological activity, only a short call from a potential fish with main energy below 1,000 Hz (CS1) and a few instances of abiotic sounds consisting of moving sediment and cavitation, and birds singing at distance.

The spectral profiles were similar across natural habitats, with most sound energy concentrated below 1,000 Hz and a gradual decline towards higher frequencies (Fig. 3). Specifically, CS1 showed a distinct spectral peak around 500–1,000 Hz, while CS2 exhibited a less pronounced energy peak around 150 Hz, consisting mostly of moving sediment and cavitation sounds, respectively (Fig. 3).

Auditory sensitivity

The Siamese fighting fish exhibited an overall auditory sensitivity bandwidth of 100–6,000 Hz. The auditory thresholds were consistently the lowest at 100 Hz (84.5 dB ±5.2 dB, mean ± standard deviation; 75–95 dB range) and the highest at 4,000 Hz (137.8 ± 3.6 dB; 130–140 range). Since auditory responses at 4,000 and 6,000 Hz were not consistently present in all recordings, they were excluded from statistical analysis. Auditory thresholds varied significantly with the frequency tested (F(6,54) =56.694, p < 0.001), and sex-specific differences were also identified (F(1,9) = 10.123, p = 0.011), with males showing lower thresholds at 400 Hz (p = 0.015) and 600 Hz (p = 0.009) compared to females. Such sex-specific differences between males and females consisted of 11.9 dB ± 3.7 at 400 Hz and 12.5 dB ± 3.8 at 600 Hz (Fig. 5).

Figure 5 Mean auditory thresholds of males and females B. splendens.

Comparison of auditory thresholds (±standard error of the mean or SEM) of males (red line) and females (grey line) B. splendens (F(1,9) = 10.123, p = 0.011). Asterisks indicate sex-specific differences at 400 Hz (p = 0.015) and 600 Hz (p = 0.009). Data presented is based on 8 months old fish, consisting of similar sized individuals–five males and six females. A wild type specimen from Chiang Rai, Thailand is shown (top).

Furthermore, the species’ auditory thresholds were above the spectral profiles of all natural soundscapes at least 15 dB, indicating minimal possibility for auditory masking due to overall background noise (Fig. 6).

Figure 6 Sound spectra from the natural habitats of B. splendens and mean auditory thresholds from the study species and a sympatric vocal fish croaking gourami Trichopsis vittata.

Sound spectra from the natural habitats of B. splendens (sampling frequency 48 kHz, FFT size 2048) and mean auditory thresholds from the study species and a sympatric vocal fish croaking gourami Trichopsis vittata (source: Wysocki & Ladich, 2001). Grey line represents a spectrum of a croaking sound from T. vittata recorded at 25° C (provided by F. Ladich, sampling frequency 44.1 kHz, bypass filtered between 0.2−3.5 kHz, FFT size 2048). Drawing of T. vittata based on Ladich (2007). Below the drawing an oscillogram of three croaking sounds is presented.

We further compared these findings with the audiogram and sound spectra from vocalizations emitted by T. vittata (Fig. 6). This soniferous fish exhibited lower auditory thresholds at frequencies matching the main spectral energy of its vocalizations within 800–2,000 Hz. This contrasted with B. splendens, which demonstrated highest sensitivity at lower frequencies between 100 and 400 Hz.

Discussion

The present work provides the first characterization of the underwater soundscape in the natural habitats of the Siamese fighting fish B. splendens, specifically in quiet, lentic freshwater systems within marshlands in Chiang Rai, Northern Thailand. In addition, we present the first description of the auditory thresholds of B. splendens, showing sex-specific differences and higher sensitivity to low frequencies compared to a sympatric vocal species, T. vittata. Overall, this study establishes an important foundation for future research on the sensory adaptation of this air-breathing anabantoid species, which is renowned for its significance in the ornamental fish market and increasingly valued for genetic and behavioural research.

Freshwater soundscapes

Despite the critical role of freshwater ecosystems in sustaining fish diversity, limited information is available on essential ecological parameters such as soundscape levels and composition (Tonolla et al., 2010; Desjonquères et al., 2015; Lara & Vasconcelos, 2019; Rountree, Juanes & Bolgan, 2020; Decker et al., 2020).

The existing studies on habitat characterization of B. splendens have primarily focused on typical ecological features such as water quality (Nur et al., 2022). According to these studies, the species inhabits water bodies with low oxygen levels and muddy substrates, which were also identified in the marshland habitats investigated in the present study. Our recorded temperatures (22–25 °C) were lower than those reported in previous studies that can vary up to 30 °C (Jaroensutansinee & Jaroensutansinee, 2005), which could be attributed to differences in geographical location, elevation, or seasonal variations. The use of dataloggers or systematic data collection would be necessary to confirm if indeed the temperature is generally lower in this region of Thailand.

Freshwater habitats can be considerably variable and rich in acoustic information providing important cues for fish orientation and survival. Noise levels in these environments depend on factors such as water flow strength and substrate composition. Lakes and backwaters typically exhibit lower noise levels compared to the fast-flowing waters found in streams and rivers, with noise levels that can differ by over 40 dB (Amoser & Ladich, 2005; Amoser & Ladich, 2010; Wysocki, Amoser & Ladich, 2007; Lara & Vasconcelos, 2019). Our results revealed relatively quiet soundscapes with SPL around 102–105 dB re 1 µPa and low variability. Lara & Vasconcelos (2019) reported similar values of 103–107 dB re 1 µPa in shallow pools and low-flow water courses with various substrates (bedrock, sand, silt; maximum 50 cm depth) in Southwest India. In Austria, Wysocki, Amoser & Ladich (2007) measured 99, 98, and 110 dB re 1 µPa, for backwaters, pond and stream with bedrock substrate, respectively.

Regarding noise spectral profiles, freshwater ecosystems such as ponds, small streams and rivers typically show higher spectral energy at the lowest frequencies followed by a gradual noise decline towards higher frequencies (Amoser & Ladich, 2005; Amoser & Ladich, 2010; Wysocki, Amoser & Ladich, 2007; Lara & Vasconcelos, 2019). We also found a similar pattern of energy decline with increasing frequency in all study sites, with most spectral energy below 1,000 Hz. Since all habitats investigated were relatively quiet, there was no distinct silent window related to sound propagation in the shallow environment, as found in previous studies (Lugli, 2010; Lara & Vasconcelos, 2019).

Our sound recordings captured diverse biological sounds, including potential fish vocalizations (see Amorim, 2006 for fish sound variability), but primarily insect sounds. Such biophony has been described in prior studies of different freshwater habitats, such as ponds and rivers (Anderson, Rountree & Juanes, 2008; Montie, Vega & Powell, 2015; Desjonquères et al., 2020; Greenhalgh, Genner & Jones, 2023). Interestingly, in several of the study sites, we observed individuals of the croaking gourami T. vittata. Although we cannot confirm the sound source, the pulsed temporal patterns in the sound recording (Fig. 3C) suggest they might consist of calls from T. vittate (e.g., Ladich, 2007). Future research should include long-term recordings and visual censuses to accurately identify soniferous fish species in these marshland habitats. Additionally, occasional abiotic sounds were also detected, such as cavitationing and moving substrate, consistent with other studies focusing on either lentic or fast-flow water courses (Tonolla et al., 2011; Lara & Vasconcelos, 2019).

An important aspect concerning the quiet nature of the marshland habitats investigated. Shallow and lentic freshwater ecosystems are known to be relatively quiet compared to watercourses with higher hydrodynamics. These shallow habitats are particularly vulnerable to climate change, overexploitation, water pollution, habitat destruction, and invasion by exotic species (Dudgeon et al., 2006), including in Thailand (Pomoim et al., 2022). Additional environmental stressors such as noise pollution could pose significant threats to the adaptation of fish species and ultimately impact community structure and ecosystem health.

Auditory sensory adaptation

We present the first description of the auditory thresholds of B. splendens, which revealed an overall sensitivity bandwidth of 100–6,000 Hz. The auditory thresholds were consistently the lowest at 100 Hz (85 dB mean, 75–95 dB range) and a gradual increment was observed towards higher frequencies, with highest thresholds recorded at 4,000 Hz (138 dB, 130–140 dB range). Our findings are consistent with previously reported audiograms for other anabantoid species, which also showed enhanced sound-detecting abilities up to 5 kHz, the highest frequency tested (Ladich & Yan, 1998). However, the best hearing sensitivities seem to vary considerably across anabantoids. Ladich & Yan (1998) measured AEP recordings from five different species, including three vocal—Trichopsis vittata, T. pumila, Colisa lalia, and two non-vocal—Macropodus opercularis and Trichogaster trichopterus. The authors identified a high-frequency sensitivity peak between 800 Hz and 1,500 Hz for all species (Ladich & Yan, 1998). This sensitivity peak matched the dominant frequencies of the vocalizations in T. vittata (1,000–2,000 Hz) and C. lalia (800–1,000 Hz), but not in the smallest species T. pumila that revealed best hearing below 1,500 Hz and most sound energy at 1,500–2,500 Hz. According to Ladich & Yan (1998), in vocal anabantoids, the association between high-pitched sounds and enhanced hearing may be attributed to the suprabranchial air-breathing chamber. Positioned near both the auditory and sonic organs (pectoral fin tendons), this chamber is known to enhance auditory sensitivity (Schneider, 1942; Yan, 1998) and likely sound production at its resonant frequency (Ladich & Yan, 1998).

In our study, B. splendens demonstrated best auditory sensitivity in the 100–400 Hz range, with mean thresholds around 85–87 dB, which were notably lower compared to those observed in other anabantoids within this frequency range. Following recordings carried in our lab (personal observations), B. splendens does not seem to produce vocalizations and lacks sonic organs (Ladich & Popper, 2001). Thus, differences in hearing curves between the sympatric B. splendens and T. vittata may relate to the absence of acoustic signalling in the former, although further research is needed to verify this hypothesis. Nevertheless, caution is needed when comparing auditory thresholds across different laboratories, as variations in experimental setups and methods of threshold determination may lead to significant differences between species at specific frequencies (Hawkins, 1981; Ladich & Fay, 2013; Maruska & Sisneros, 2016).

The present study also suggests that anabantoids, including both B. splendens and T. vittata, are well adapted to the ambient noise conditions typical of their shallow, lentic, marshland freshwater habitats. Our findings indicate that neither species appears to be masked by the background noise present in their environments. Eco-acoustical constraints likely account for the diversity in fish hearing sensitivities (Ladich & Schulz-Mirbach, 2016). In the case of anabantoids, the development of the suprabranchial air-breathing chamber serves not only as an adaptation to low oxygen environments but also to enhance hearing sensitivity under low ambient noise conditions. This enables fish to detect subtle abiotic acoustic cues as well as sounds from conspecifics and/or heterospecifics, including predators and prey.

Furthermore, we identified sex-specific differences in B. splendens, with males exhibiting lower thresholds, approximately 12–13 dB lower at 400 and 600 Hz compared to females. This finding represents the first evidence of such differences among anabantoids and suggests a possible correlation with variations in the suprabranchial air-breathing chamber, although other hypothesis can be considered. Males of this species are known for their pronounced aggression and higher oxygen demands, showing significantly more air breathing compared to females (Castro et al., 2006; Alton, Portugal & White, 2013). Such sex-specific differences may have influenced the development and morphology of the suprabranchial chamber, potentially affecting auditory sensitivity (Yan, 1998; Ladich & Popper, 2004). Moreover, several studies focusing on soniferous fish have shown that variations in gonadal state, circulating sex-steroids, and steroid receptor expression in peripheral and central auditory structures are associated with changes in auditory sensitivity and inner ear hair cell count (reviewed in Maruska & Sisneros (2015)). The sensory enhancement in males, which are particularly aggressive and provide parental care (Jaroensutansinee & Jaroensutansinee, 2005; Castro et al., 2006) may offer advantages in detecting acoustic cues from conspecifics, predators and prey.

Future research should explore potential sex-specific differences in other anabantoid species, focusing on whether the air-breathing organ and inner ear sensory epithelia differ between males and females. Additionally, it should investigate the potential influence of steroids, which are known to influence aggression levels in males, on auditory thresholds of the study species.

In conclusion, the present work shows that B. splendens hearing range is best at low frequencies, progressively decaying towards higher frequencies. Interestingly, consistent differences in hearing sensitivity between males and females were detected, calling for further studies on the possible function and causes of this variability. The difference in auditory sensitivity to the closely related and sympatric vocal species T. vittata, supports the hypothesis that conspecific vocal communication is a significant selective pressure acting on auditory systems. Lastly, the study shows that natural habitats of B. splendens are relatively silent, highlighting the need to monitor and preserve soundscapes in conservation programs of freshwater habitats.

Supplemental Information

Supplemental Information 1 Auditory thresholds based on Auditory Evoked potentials

Auditory thresholds of 5 males and 6 females B. splendens. Data presented is based on 8 months old fish, consisting of similar sized individuals.

Supplemental Information 2 Sound recording sample from RT2 field site

Example of sound recording (5min) from Rai Son Thon Reservoir (RST2) presenting biological activity (eg. high-pitched insect calls) and cavitation sounds. The ambient sounds were recorded using a hydrophone (Brüel & Kjær 8104) connected to a hand-held sound level meter with sound recording function (Brüel & Kjær 2250). Sampling frequency 48 kHz.

Supplemental Information 3 Example of sound recording with insect calls and potential fish vocalizations in RT2 site

Example of sound recording from Rai Son Thon Reservoir (RST2) with insect calls and potential fish vocalizations. Sound recorded using a hydrophone (Brüel & Kjær 8104) connected to a hand-held sound level meter with sound recording function (Brüel & Kjær 2250). Sampling frequency 48 kHz.

Supplemental Information 4 Example of sound recording from fish in RT3 site

Example of sound recording from Rai Son Thon Reservoir (RST3) from a possible fish species. Sound recorded using a hydrophone (Brüel & Kjær 8104) connected to a hand-held sound level meter with sound recording function (Brüel & Kjær 2250). Sampling frequency 48 kHz, high-pass filtered at 0.4 kHz.

Supplemental Information 5 ARRIVE checklist

We are thankful to Precha Jintasaerewonge for the support provided in finding the locations and communicating with local people during the field work in Thailand. We thank Friedrich Ladich for providing the sound recording of the croaking gourami T. vittata.

Additional Information and Declarations

Competing Interests

Author Contributions

Animal Ethics

Field Study Permissions

Data Availability

The authors declare there are no competing interests.

Andreia Ramos performed the experiments, analyzed the data, prepared figures and/or tables, authored or reviewed drafts of the article, and approved the final draft.

David Gonçalves analyzed the data, authored or reviewed drafts of the article, and approved the final draft.

Raquel O. Vasconcelos conceived and designed the experiments, analyzed the data, prepared figures and/or tables, authored or reviewed drafts of the article, and approved the final draft.

The following information was supplied relating to ethical approvals (i.e., approving body and any reference numbers):

All experimental procedures were approved by the Division of Animal Control and Inspection of the Civic and Municipal Affairs Bureau of Macao, license AL017/DICV/SIS/2016.

The following information was supplied relating to field study approvals (i.e., approving body and any reference numbers):

No permission required as no sampling of animals was conducted.

The following information was supplied regarding data availability:

The raw measurements, auditory thresholds of Betta splendens - males and females and examples of original field sound recordings are available in the Supplementary Files.

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
