# Peer review of "Exploring freshwater soundscapes of tropical marshland habitats in Southeast Asia: insights into auditory sensory adaptation of wild Siamese fighting fish Betta splendens"

_PeerJ, doi:10.7717/peerj.18491_

## Round 0.1 · original submission · Minor Revisions

The manuscript has a high level of writing quality, with clearly delineated research objectives, methodologies, and findings. Although the experimental design, ethical considerations, and statistical analyses appear to be appropriate, there are some recommendations regarding the elucidation of methodological procedures, and the visual representation of results are provided herein.

Reviewer 1 ·

Basic reporting

Please see my comments in the review.

Experimental design

Please see my comments in the review.

Validity of the findings

Please see my comments in the review..

Additional comments

Please see my comments in the review..

Annotated reviews are not available for download in order to protect the identity of reviewers who chose to remain anonymous.

·

Basic reporting

Fine

Experimental design

Does the job.

Validity of the findings

Findings valid

Additional comments

Ramos et al present a well-done manuscript categorizing background noise in a number of small Asian streams with non or slow-flowing water. They also use evoked potentials to measure hearing in a stock of Siamese fighting fish (Betta splendens) derived from their native habitats and thus relate acoustic conditions to the evolution of hearing. There is very little auditory work that relates to natural sound spectra, making this a novel contribution. Interestingly, they find a major difference in auditory thresholds between male and female Bettas, which is also novel and currently unexplained. Background noise is considerably below auditory thresholds, at least under sunny conditions and thus does not appear to be a major factor in the evolution of hearing, which could potentially be affected historically by storms at other seasons?
They also utilize work from Fritz Ladich’s lab on croaking gourami sounds and hearing since both species originate from the same geographical areas. There is a wide difference in hearing curves between the two species, which correlates with sound production in the gouramis compared to mute Bettas, another interesting finding since they are from the same family.
The paper is well written although aspects of the introduction and discussion could be shortened. This is not a major criticism. The figures are lovely.
A few minor comments
L 125. Write out LZS.
L 138. Somewhere in the paragraph include the dates of recording and approximate times since both sound spectra and water quality measures could change seasonally or even daily.
L 165. Re: All sound recordings were performed away from human populations, hence no ethical concerns related to accidentally recording human activity were present.

This statement about ethics is plain silly. I don’t know if the authors are politically hip or someone is compelling them to write this sort of drivel. Since sticking a hydrophone in water even in a populated area is unlikely to disturb anyone.

L 214. Replace to generate with generation of.

L 247. Capitalize the S for Siemens.

L 333. Vittate. Correct spelling.

L 377. Re: gender differences. This represents another intrusion of political correctness. Since contemporary progressives have come up with something like 48 different genders based on personal preferences of humans, the word has no business in a biology paper. Please use the word sex or sexual. I realize there may be intersex fishes and ontogenetic sex changes, but in general fish are referred to by their sex as male or female. Note both of my political suggestions are minor and do not affect the message of the paper.

Fig. 3 legend. Include the dates of recording since there may be other spectra during the rainy season.

Reviewer 3 ·

Basic reporting

A. Clear, unambiguous, and professional English is used throughout.
Yes.
B. Introduction and Background to show context.
Yes.

C. Literature referenced & relevant.
The paper citation is not up to date.
Some significant articles pertinent to this study should be included. Here are a few examples:
1. Ladich, F. and R. R. Fay. 2013. Auditory evoked potential audiometry in fish.
Rev Fish Biol Fish, 23(3):317-364. doi: 10.1007/s11160-012-9297-z.
2. Ladich, F. and Schulz-Mirbach, T. 2016. Diversity in fish auditory systems: one of the riddles of sensory biology. Front. Ecol. Evol., 31.
3. Saidel, W.M. and A.N. Popper 1987. Sound perception in two anabantid fishes. Comp. Biochem. Phsiol., 88A (1): 37-44.
4. Yan, H.Y. 1998. Auditory role of the suprabranchial chamber in gourami fish. J. Comp. Physiol. A, 183: 325-333.

D. The structure adheres to PeerJ standards, discipline norm, or improved for clarity.
Figures are relevant, high-quality, well-labeled, and thoroughly described.
Fig. 2C should be enlarged.
E. Raw data has been supplied.
Yes.

Experimental design

A. Original primary research within Scope of the journal.
Yes.
B. The research question is well-defined, relevant, and meaningful. The research fills an identified knowledge gap.
Yes or no? The audiogram of Betta splendens has not yet been measured; therefore, there is a need to conduct this research. However, this topic is not innovative and does not effectively address a well identified knowledge gap. The authors did not mention what has led them to expect there might be sexual difference in hearing sensitivity (e.g., Line 388-389: Surprisingly, consistent differences in hearing sensitivity between males and females were detected), and I guess it is just an exploratory research.
There are published works on the soundscapes of freshwater ecosystems, therefore, the related part in this study is not innovative.

C. A rigorous investigation was conducted to uphold high technical and ethical standards.
Yes.
D. The methods are described with sufficient detail and information to allow for replication.
Yes.

Validity of the findings

A. Meaningful replication is encouraged when the rationale and benefits to literature are clearly articulated.
Yes.
B. All underlying data have been provided; they are robust, statistically sound, and controlled.
Line 234 to 236:”Parametric assumptions were complied, namely data were normally distributed and variances were homogeneous.” Why the authors did not test if the data set were normally distributed?
C. The conclusions are clearly articulated, directly connected to original research question & limited to supporting results.
Yes.

Additional comments

General comments:
1. Title: "..freshwater soundscapes.." lacks specificity. It is recommended to modify it, for example, to "..marshland underwater soundscapes.."
2. The present study demonstrates that Betta splendens hearing range is best at low frequencies and consistent differences in hearing sensitivity between males and females were detected. The study also indicates that the natural habitats of B. splendens are relatively silent, highlighting the attention to document and preserve soundscapes in conservation programs of freshwater habitats.
The sex-specific differences in auditory sensitivity have not been explored in other anabantoids. This study introduces a new topic for future research.
3. Language and grammar meet professional standards.

Specific comments:
1. L. 29: “ecological parameters” changed to” hydrological parameters”.
2. L. 34. “potential fish vocalization”. Please provide the characteristics of fish sounds along with relevant citations.
3. Lines 254-257 contain content that is repeated in lines 258-261.
4. Lines 254-257:“Soundscapes from RST locations revealed distinct biological sounds, including potential fish vocalizations: series of pulsed sounds < 2000 Hz (RST2) and short call with main energy below 1000 Hz (RST3); birds singing at distance with main energy between 1000-7000 Hz (RST2), but primarily insect sounds >2000 Hz, especially in RST2”.
It is recommended that references on underwater insect sounds and fish sounds in the riverine system, in terms of spectral density, should be given.
Regarding bird songs at a distance, how much acoustic energy can be expected from these sounds after they enter the water?
5. Line 366: "range.." Please remove the additional full stop.
6. Line 371: “we identified sex-specific differences in B. splendens, with males exhibiting lower thresholds, approximately 12-13 dB lower at 400 and 600 Hz compared to females. This finding represents the first evidence of such differences among anabantoids and suggests a possible correlation with variations in the suprabranchial air-breathing chamber.”
The authors considered the possible correlation that the sex-specific difference is related to variation in the suprabranchial air-breathing chamber and it represents a significant hypothesis. Information about the structure in both sexes should better be present in this study.
It is suggested that the authors should also explore the possible benefit to male when they are more sensitive to sounds at 100 – 400 Hz than female; increase of fitness can be a possible mechanism that enhance the differentiation of such sex-specific difference.
It is known that Betta splendens, particularly the males, blow gas bubbles to construct a floating bubble nest and get ready to mate. During construction of the bubble nest or after it is constructed, whether or not there are specific noises generated associated with the bubbles deserved to be observed. These noises might become an acoustical communicatory signals which induced the differentiation of such sex-specific difference. This is just my personal imagination.
7. Lines 364-366: “In our study, B. splendens demonstrated best auditory sensitivity in the 100-400 Hz range, with thresholds around 85-87 dB, which were notably lower compared to those observed in other anabantoids within this frequency range. “
Saidel and Popper (1987; which is not cited in the present manuscript) showed that “Saccular microphonics were recorded in both species (Trichogaster trichopterus and Helostoma temincki) from 80 to 1600Hz, with lowest thresholds between 100 and 200 Hz. The overall microphonic response curves (sensitivity and bandwidth) of the two species were statistically similar to one another with an analysis of variance, although there were statistically different thresholds at 100 and 800 Hz.” Although the experimental methods differ between the present study and Saidel and Poper, both studies indicate higher mechanical sensitivity at the low frequency range in these anabantoid species.
8. Line 383: “Future research should explore potential sex-specific distinctions in the air-breathing organ and inner ear sensory epithelia, as well as the potential influence of steroids, which are known to influence aggression levels in males, on auditory thresholds of the study species.”
It is recommended that future research investigate whether other anabantoid species showing such sex-specific difference in auditory sensitivity. If there are additional examples, what common biological characters that might lead to such sex-specific difference.
9. Lines 390-392: ‘The difference in auditory sensitivity to the closely related and sympatric vocal species T. vittata, supports the hypothesis that conspecific vocal communication is a significant selective pressure acting on auditory systems.’ It means that the reproductive or advertisement calls of T. vittata leads to coupling of the auditory sensitivity. While Betta splendens do not vocally communicate during courting or reproduction, such coupling becomes unnecessary. However, it does not explain the reason for the sex-specific difference in hearing sensitivity.

Despite that the finding on sex-specific difference in hearing sensitivity is new, the authors have not provided further persuasive speculation on what might cause such sexual difference relating to the fitness of having a higher sensitive at low frequencies in male. Personally, I do not believe the impact and novelty of this study are very high.

Reviewer 4 ·

Basic reporting

Clearly writting with clear research aims. Well referenced.

Experimental design

Determining the auditory thresholds of fish is outside the scope of my expertise but the experimental design, ethical concerns, and statistical analysis seem appropriate to me and in-line with similar research in the field.

Methods are very well described throughout.

Validity of the findings

The results and data are clearly presented and radily avaliable.

Annotated reviews are not available for download in order to protect the identity of reviewers who chose to remain anonymous.

---

## Round 0.2 · accepted · Accept

Thank you for submitting the new version of your manuscript. After checking that all the suggestions sent by the reviewers have been taken into account and have improved your manuscript, I am pleased to inform you that your work can be accepted for publication.

Thank you for submitting your work to this journal.